# Development of Mass Production Technology of Highly Permeable Nano-Porous Supports for Silica-Based Separation Membranes

**DOI:** 10.3390/membranes9080103

**Published:** 2019-08-16

**Authors:** Ken-ichi Sawamura, Shigeru Okamoto, Yoshihiro Todokoro

**Affiliations:** eSep Inc., Keihanna Open Innovation Center @ Kyoto (KICK),1-5-7 Seikadai, Seika-cho, Souraku-gun, Kyoto 619-0238, Japan

**Keywords:** gas separation, pore size control, molecular sieve

## Abstract

Silica-based membranes show both robust properties and high-permeability, offering us great potential for applying them to harsh conditions where conventional organic membranes cannot work. Despite the increasing number of paper and patents of silica-based membranes, their industrial applications have yet to be fully realized, possibly due to their lack of technologies on scaling-up and mass production. In particular, quality of membrane supports decisively impacts final quality of silica-based separation membranes. In this study, therefore, we have developed mass producing technologies of nano-porous supports (φ 12 mm, length 400 mm) with surface center pore size distribution of 1–10 nm, which are generally used as supports for preparing separation membranes with a pore size of less than 1 nm. The developed mass production apparatuses have enabled us to reproducibly produce nano-porous silica-based supports with high permeance (e.g., N_2_ permeance of more than 10^−5^ mol m^−2^ s^−1^·Pa^−1^) minimizing effects of membrane defects less than 0.1% of the total flux. The developed nano-porous supports have enabled us to reproducibly produce silica-based separation membranes with high permeace and selectivity (e.g., H_2_ permeance of about 5 × 10^−6^ mol m^−2^ s^−1^ Pa^−1^ and H_2_/SF_6_ permeance ratio of more than 2000).

## 1. Introduction

Membrane-based separation is one of the promising technologies for simplifying processes and reducing energy consumption drastically in future chemical and petroleum industries. Silica-based membranes, such as SiO_2_ membranes [1], SiO_2_–ZrO_2_ membranes [2], or organo-silica membranes [3,4,5], show both robust properties and high-permeability, offering us great potential for applying them to harsh conditions where conventional organic membranes cannot work.

Despite the increasing number of papers and patents of silica-based membranes, their industrial applications have yet to be fully realized, possibly due to lack of technologies on scaling-up and mass production. In particular, the quality of membrane supports decisively impacts final quality of silica-based separation membranes. Considering industrial applications and their economic feasibility of ceramic-based separation membranes, the permeance of target separation molecules are expected to be more than 10^−6^ mol m^−2^ s^−1^ Pa^−1^ in most cases, such as hydrogen separation and water separation. Generally, silica-based molecular sieve separation layers with their pore size of less than 1 nm are formed on the surface or inside of nano-porous supports with a pore size of 1–10 nm. In order to obtain high-flux (more than 10^−6^ mol m^−2^ s^−1^ Pa^−1^) and high-selective membrane with their pore size of less than 1 nm, the permeances of the nano-porous supports with their pore size distribution of 1–10 nm need to be more than 10^−5^ mol m^−2^ s^−1^ Pa^−1^. In addition, effects of permeation pathway through defects, such as pinholes or cracks of the nano-porous supports should be minimized to obtain reasonable separation performance after membrane separation layer coated.

The purpose of this study is to develop mass producing technologies of industrially applicable, highly permeable, and defect-free nano-porous supports for producing silica-based membranes. In this study, firstly, highly permeable silica-based nano-porous supports were developed by controlling size of nano-particles deposited on the surface of the supports. Secondly, an example of our mass-producing apparatus for the nano-porous supports was introduced. Finally, the developed nano-porous supports were tested to prepared silica-based separation membranes for hydrogen separation.

## 2. Materials and Methods

In order to effectively develop technologies on scaling-up and mass production of silica-based supports and membranes, we focused on clarifying correlations among synthesis conditions, structural properties, and gas permeation properties. Materials and methods of synthesis, structural characterization, and gas permeation tests are shown below.

### 2.1. Synthesis

Silica-based nano-porous supports and membranes were formed on α-alumina symmetric porous tubes (average pore size: 3 µm, outside diameter ϕ: 12 mm, length: 400 mm, IWAO JIKI KYOGYO Co., Ltd., Saga prefecture, Japan) by sol-gel method, based on the procedure reported by the research group of Hiroshima University [1,2,3,4,5]. First, α-alumina particles (average size: 0.2 µm) were coated on the α-alumina porous support to obtain a smooth surface layer (α-alumina intermediate layer). Next, silica-based intermediate layers were formed on the surface of the α-alumina intermediate layer, using colloidal sols of SiO_2_ [1], SiO_2_–ZrO_2_ (Si/Zr = 1) [2], or 1,2-bis(triethoxysilyl)ethane (BTESE)-derived organo-silica [3,4,5]. The colloidal sols were prepared by hydrolysis and condensation reactions of tetra-ethoxysilene (TEOS), zirconium tetra n-butoxide, or BTESE. The detail preparation procedures of colloidal sols were reported elsewhere [1,2,3,4,5]. The particle sizes of the colloidal sols were controlled by adjusting sol concentration (1–2 wt %), aging time (1–30 days), and aging temperature (25–40 °C). 

In this study, the hot coating method [1] was employed for quick formation of a thin active top layer on the substrate. Colloidal sols of SiO_2_ or SiO_2_–ZrO_2_ were coated on the α-alumina intermediate layers at 180 °C and fired at 550 °C repeatedly for synthesizing silica-based nano-porous supports. BTESE-derived organo-silica membranes were synthesized based on the procedure reported by the research group of Hiroshima University [5]. Some experimental conditions were modified to improve productivity, as mentioned below. Colloidal sols of BTESE-derived organo-silica (0.5 wt % in ethanol) were coated on the SiO_2_–ZrO_2_ nano-porous supports at 145 °C with a piece of wet cloth and fired at 350 °C for 7 min in N_2_ repeatedly for synthesizing gas separation membranes. This procedure was repeated 4 times to obtain BTESE membranes with less defects. The SiO_2_–ZrO_2_ nano-porous supports used for synthesizing the BTESE membranes had high permeance (e.g., N_2_ permeance of more than 10–5 mol m^−2^ s^−1^ Pa^−1^) with an average pore size of about 1–2 nm, as shown later. 

Figure 1 shows two types of experimental apparatuses (batch-type and flow-type) for synthesizing silica-based nano-porous supports and membranes. In this study, the batch-type apparatus was used for obtaining fundamental experimental data. In the batch-type apparatus, first, the substrates were preheated in a heating oven at 145 or 180 °C for 4 min. Next, each colloidal sol was coated on the surface of preheated substrates by contacting it with a piece of wet cloth. Then, the supports or membranes were directly inserted in a furnace at 350 or 550 °C and kept there for 7–15 min. After the firing, the supports or membranes were naturally cooled for 4–10 min. These procedures were manually repeated 4–10 times with each colloidal sol prepared. In case of the batch-type apparatus in Figure 1, average productivity of the hot coating of each colloidal sol was limited to 10 samples per unit furnace. 

In order to enhance the productivity, a flow-type apparatus was prototyped. In the flow-type apparatus, substrates samples are designed to be circulated from preheating chamber at 180 °C for 10 min → hot coating device → firing chamber at 550 °C for 10 min → cooling chamber for 4–6 min. The residence times in each of the chambers were controlled by adjusting the feeding speed of belt conveyors. The flow-type apparatus has enabled the coating cycle to repeat continuously and automatically. Compared with the batch-type apparatus (manual operation), the flow-type apparatus (automatic operation) in Figure 1 improved the productivity from 10 to 60 samples per unit apparatus. For example, 6 times coating of 60 samples with one kind of colloidal sol will be completed within 3 h using the flow-type apparatus. In case of the flow-type apparatus, its production capacity will be easily expanded by increasing the number of membrane cartridge and prolong the line of electric furnaces.

### 2.2. Structural Characterization

The sizes of colloidal sols of SiO_2_, SiO_2_–ZrO_2_, and organo-silica synthesized were measured by a dynamic light scattering (DLS) analyzer (Malvern Zetasizer Nano ZS, Worcestershire, England) and a transmission electron microscope (TEM). The structures of synthesized silica-based nano-porous supports were observed by a scanning electron microscope (SEM). 

### 2.3. Gas Permeation Test

Two types of gas permeation tests, a nano-permporometory [6,7] and single gas permeation tests, were employed to evaluate quality of synthesized silica-based nano-porous supports and membranes.

The synthesized silica-based nano-porous supports were evaluated by a nano-permporometory, which is suitable for investigating pore size distribution in the range of 1–10 nm. The synthesized silica-based membranes were evaluated by single-gas permeation tests, which are suitable for investigating membranes with their pore size of less than 1 nm. 

In the gas permeation tests, as-made nano-porous supports or membranes (outside diameter ϕ: 12 mm, length: 400 mm) were placed inside the module cell and sealed with silicon caps at both ends. The effective membrane was 143 cm^2^. 

#### 2.3.1. Nano-Permporometory

The pore size distribution of synthesized silica-based nano-porous supports was evaluated by a nano-permporometory. In this method, non-condensable gas permeances through a porous sample are measured with increasing relative pressure of condensable vapor in a stepwise manner. The contributions of gas permeations via a specific size of pores are estimated by blocking their nanopores with a vapor condensate. The diameters of the pores blocked by the condensate as a function of vapor partial pressure are estimated using the well-known Kelvin Equation (1)
D_K_ = −4υσ cosθ/RT ln(P/P_s_)(1)
where D_K_, υ, σ, and θ are Kelvin diameter, molar volume, surface tension, and contact angle, respectively. In this study, N_2_ (non-condensable gas) and water (condensable vapor) were used, assuming cosθ = 1 in calculation of Kelvin diameter of SiO_2_ and SiO_2_–ZrO_2_-based nano-porous supports because both supports were hydrophilic, probably due to immediate formations of –OH at surface of the supports. The nano-permporometry measurements were performed at 25 °C.

The schematic diagram of the nano-permporometry measurements and their experimental conditions are shown in Figure 2 and Table 1, respectively. Samples were attached in a glass module and sealed with silicon caps as shown in Figure 2. 

#### 2.3.2. Single-Gas Permeation Test

Single-gas permeation tests were conducted after membranes were preheated at 350 °C in N_2_ to remove chemisorbed water molecules. Samples were attached in a glass module and sealed with silicon caps. Pure gases of H_2_ (0.29 nm), CH_4_ (0.38 nm), or SF_6_ (0.55 nm) were fed to the outside of a cylindrical membrane at 4–20 kPa. The gas permeation area was kept at 100 °C by a thermostatic chamber. The permeance of gases was measured by a bubble film meter.

## 3. Results

In this section, results of characterization of coating nano-particles, synthesized silica-based nano-porous supports, and membranes are shown in this order.

### 3.1. Characterization of Coating Nano-Particles

Table 2 summarizes estimated particle sizes of colloidal sols prepared with different synthesis conditions. The particle sizes of colloidal sols increased with increasing sol concentration, aging time, or aging temperature. TEM offers decisive evidence for determining the size of nano-particles when their clear images are obtained. Since relatively clear TEM images were obtained in Sample-1, Sample-2, Sample-4, and Sample-6, their TEM images are shown in Figure 3. Because clear TEM images of the other samples were not obtained in this study, their particle sizes are estimated by DLS. As shown in the Figure 3, the sizes of prepared nano-sol samples had some distributions. 

### 3.2. Characterization of Synthesized Silica-Based Nano-Porous Supports

In this section, characterization results of nano-porous supports and membranes synthesized using the batch-type apparatus (as shown in Figure 1) are explained except otherwise mentioned. The results of those synthesized using the flow-type apparatus are explained in Section 3.2.3.

#### 3.2.1. General Structure

Figure 4 shows typical SEM images of the synthesized silica-based nano-porous supports. From the SEM images, three layers, attributed to the α-alumina porous support layer, the α-alumina intermediate layer, and the silica-base layer, were observed. The α-alumina porous support layer was formed with large (more than 1 µm) alumina particles. The α-alumina intermediate layer was formed with fine (about 0.2 µm) alumina particles on the upper surface of the α-alumina porous support. The silica-based layers were formed on the upper surface of the α-alumina intermediate layer. The total thickness of the silica-based layers was less than 500 nm. The observed silica-based layer in Figure 4 was synthesized by multi-coating SiO_2_-based nano-sols of Sample-2 (5–15 nm) and Sample-4 (2–10 nm) in this order. Probably due to the limitation of resolution of the SEM, no clear boundary between the silica layers with different particle sizes were observed from the SEM observation.

#### 3.2.2. Pore Size Distribution of SiO_2_-Based Nano-Porous Supports

Nano-permporometry profiles of nano-porous supports prepared by coating nano-sol samples with different particle sizes are shown in Figure 5 (SiO_2_-based nano-porous supports). Based on the results of nano-permporometry measurements, distributions of gas permeation ratio through each of permeation path with different pore sizes were calculated and summarized in Table 3.

#### 3.2.3. Reproducibility in the Flow-Type Production

Different from the manually operative batch-type apparatus, the automatic operative flow-type apparatus seems suitable for mass production of the nano-porous supports and membranes. Figure 6 and Table 4 compare nano-permporometry profiles of nano-porous supports prepared with the flow-type apparatus prototyped in this study. The nano-sol of Sample-3 was used to prepare the nano-porous supports with the prototyped apparatus. All samples measured seem to have almost the same nano-porometry profiles with high permeance (e.g., N_2_ permeance of more than 10^−5^ mol m^−2^ s^−1^ Pa^−1^) minimizing effects of membrane defects less than 0.1% of total flux. 

#### 3.2.4. Pore Size Distribution of SiO_2_–ZrO_2_-Based Nano-Porous Supports

Compared with pure SiO_2_-based nano-porous supports, SiO_2_–ZrO_2_-based nano-porous supports are less stable in a strong acidic condition but more stable against water and alkali. Therefore, development of SiO_2_–ZrO_2_-based nano-porous supports are also indispensable for applying them to conditions where pure SiO_2_-based materials are unsuitable. SiO_2_–ZrO_2_-based nano-porous supports are often used as intermediate layers for preparing organo-silica membranes for gas separation [5], probably due to better stability against water. 

Figure 7 shows nano-permporometry profiles of SiO_2_–ZrO_2_-based nano-porous supports prepared by coating nano-sol samples with different particle sizes Based on the results of nano-permporometry measurements, distributions of gas permeation ratio through each of permeation paths with different pore sizes were calculated and summarized in Table 5. 

### 3.3. Characterization of Synthesized Organo-Silica Membranes

The BTESE-derived organo-silica membranes were synthesized by multi-coatings of nano-sols of Sample-5 (SiO_2_–ZrO_2_, 2–10 nm), Sample-6 (SiO_2_–ZrO_2_, 2–5 nm), and Sample-7 (BTESE derived organo-silica, 1.5 nm) in this order. Different from the other samples (Sample 1 to 6), Sample-7 (BTESE derived organo-silica) showed water-repellent property. Since the top surface of the organo-silica membranes seem hydrobobic with their pore size of less than 1 nm, single-gas permeation tests through the membranes were employed. The results of single-gas permeation tests are listed in Table 6.

## 4. Discussion

### 4.1. Correlations Among Synthesis Conditions, Structural Properties, and Gas Permeation Properties

Generally, coating nano-particle’s sizes and their distributions critically influence on pore size distribution of the prepared nano-porous supports. Nano-sols with uniform size of nano-particles seem best for preparing a nano-porous support with a specific pore size. In reality, however, the sizes of nano-particles of prepared nano-sols had distributions to some extent, as shown in Table 2. In case of spherical particles with uniform size, pore size of their close packing should be 1/3 of the spherical particles. As shown in Table 3, Table 4 and Table 5, pore size distributions of nano-porous supports synthesized showed relatively good correlations with the sizes of nano-particles used. Nano-porous supports with more uniform pore distribution will be synthesized with more uniform size of nano-particles in near future.

### 4.2. Quality of Nano-Porous Supports and Membranes

High-quality nano-porous supports and/or membranes should be highly-permeable with specific pore size distributions. Generally, thinner layers show higher permeability; however, thinner layers are more difficult to form defect-free. Existence of defects, such as pinholes and/or cracks form large pores, deteriorates their intended pore size distributions. From the SEM observations as shown in Figure 4, the silica-based layer prepared seems thin (less than 500 nm) with no clear defects. Since no clear (more than 100 nm) increase of layer’s thickness was observed after coating smaller nano-particles (Sample-4, 6, and 7) on nano-porous supports of Sample-2 and 5, effective thickness of the silica-based layer composed of smaller nano-particles (Sample-4, 6, and 7) would be less than 100 nm.

The developed nano-porous silica-based supports showed high permeance (e.g., N_2_ permeance of more than 10^−5^ mol m^−2^ s^−1^ Pa^−1^), minimizing effects of large pores (e.g., more than 10 nm) less than 0.1% of total flux, as shown in Figure 5, Figure 6 and Figure 7. Using the developed nano-porous silica-based supports, organo-silica membranes high permeace and selectivity (e.g., H_2_ permeance of about 5 × 10^−6^ mol m^−2^ s^−1^ Pa^−1^ and H_2_/SF_6_ permeance ratio of more than 2000) were reproducibly produce, as shown in Table 6. The membranes showed small permeation of CH_4_ (0.38 nm) and little permeation of SF_6_, the effective pore sizes of the membranes were estimated to be 0.4–0.5 nm. Since the type of organo-silica sol employed in this study was BTESE, the network structure of BTESE(–Si–C–C–Si–) would allow permeation of smaller molecules, or H_2_ (0.29 nm), N_2_ (0.36 nm), and CH_4_ (0.38 nm), and block permeation of larger molecules, SF_6_ (0.55 nm) [8]. Judging from the H_2_ permselectivity for silica-based membranes and zeolite membranes, Kanezashi et al. [5] estimated the order of average pore size as follows: TEOS-derived silica membrane [9] (up to 0.3 nm) < DDR-type zeolite membrane [10] (up to 0.45 nm) < BTESE-derived silica membrane [5] (up to 0.5 nm) < MFI-type zeolite membrane [11] (up to 0.56 nm). Therefore, BTESE-derived silica membranes are expected to apply to separation of hydrogen/organic gas mixtures such as hydrogen/propane, hydrogen/isobutene, and hydrogen/toluene.

The H_2_ permeation properties of the obtained BTESE-derived silica membranes were compared with those reported by other groups in Table 7. The permselectivity of BTESE-derived silica membranes were almost the same; however, H_2_ permeances were largely different from each other. In general, BTESE-derived silica membranes with their intermediate layer of SiO_2_–ZrO_2_ showed higher H_2_ permeance than those with their intermediate layer of γ-alumina [12,13,14], probably due to the higher permeability of SiO_2_–ZrO_2_ intermediate layer than those of γ-alumina intermediate layer. The research group of Hiroshima University used SiO_2_–ZrO_2_ intermediate layers and reported high H_2_ permeance of BTESE-derived silica membranes [5,15,16]; however, their H_2_ permeances differed by one order of magnitude. We assume the variation of H_2_ permeation is probably due to the difference in the permeability of intermediate layers synthesized.

We assume the use of highly permeable nano-porous supports greatly contributed to the high H_2_ permeanes of organo-silica (BTESE) membranes synthesized. Compared with γ-alumina intermediate layered supports [12,13], the permeability of the SiO_2_–ZrO_2_ intermediate layered supports developed in this study were more than ten times higher, resulting in more than ten times higher H_2_ permeances of BTESE-derived silica membranes. In this study, we have succeeded in scaling-up the high quality BTESE-derived organo-silica membranes reported by the research group of Hiroshima University [5] from the lab size (φ 10 mm, length up to 100 mm, employed in the research group of Hiroshima University) to an industrial size (φ 12 mm, length 400 mm).

### 4.3. Mass Producibility

Considering mass production of silica-based membranes, both sol-gel methods [1,2,3,4,5] and chemical vapor deposition (CVD) methods [17] would be candidates; however, sol-gel methods seem more cost-effective in preparing nano-porous supports because silica-based intermediate layers with controlled pore size distributions can be formed immediately by just coating and heating nano-particles with a specific size in order. As shown in Figure 6 and Table 4, silica-based nano-porous supports were reproducibly obtained by using the flow-type apparatus shown in Figure 1. In the flow-type production, operations in a typical sol-gel method, or the cycle of preheating → hot coating of nano-particles → firing, can be performed continuously and automatically. In this study, we have confirmed that mass productions of silica-based nano-porous supports were principally possible by the flow-type sol-gel method. Actual mass production of silica-based nano-porous supports and membranes will be our next research.

## 5. Conclusions

In this study, we have successfully scaled-up high quality silica-based nano-porous supports and membranes by controlling sizes of nano-particles coated. The developed nano-porous silica-based supports have showed high permeance (e.g., N_2_ permeance of more than 10^−5^ mol m^−2^ s^−1^ Pa^−1^), minimizing effects of membrane defects less than 0.1% of total flux. The developed nano-porous supports have enabled us to reproducibly produce silica-based separation membranes with high permeance and selectivity (e.g., H_2_ permeance of about 5 × 10^−6^ mol m^−2^ s^−1^ Pa^−1^ and H_2_/SF_6_ permeance ratio of more than 2000). In additions, a flow-type sol-gel method was developed for the mass productions of silica-based nano-porous supports and membranes.

## Figures and Tables

**Figure 1 membranes-09-00103-f001:**
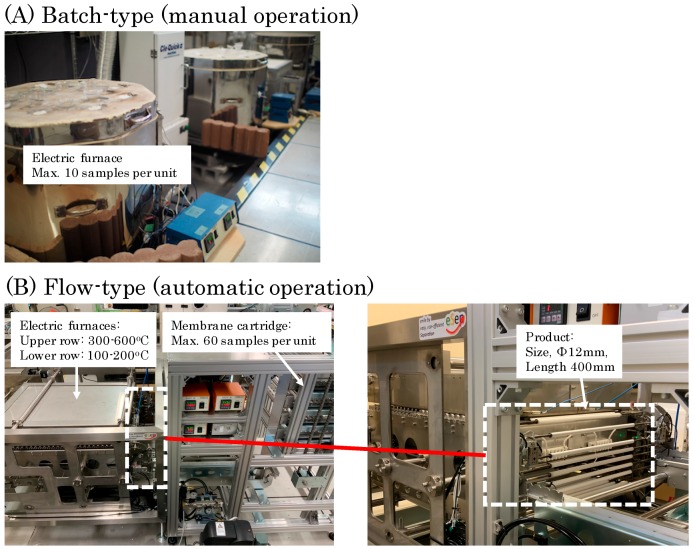
Experimental apparatuses for synthesizing silica-based nano-porous supports and membranes used in this study: (**A**) Batch-type and (**B**) flow-type apparatuses.

**Figure 2 membranes-09-00103-f002:**
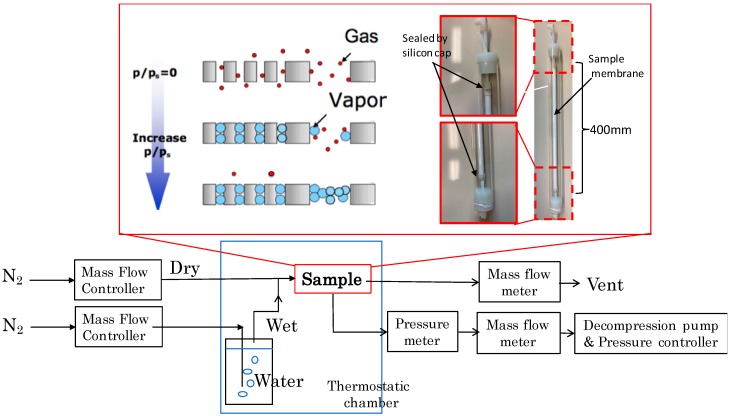
The schematic diagram of nano-permporometry measurements employed in this study.

**Figure 3 membranes-09-00103-f003:**
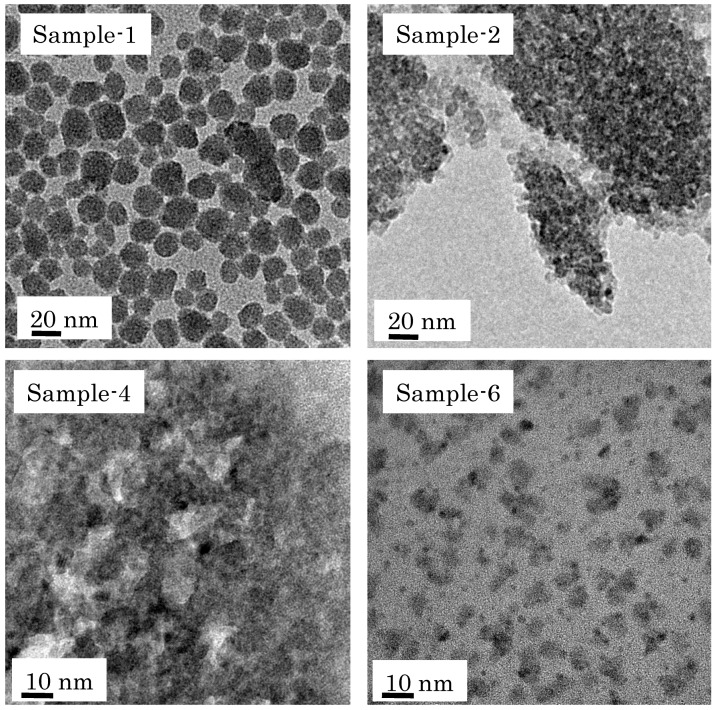
TEM images of nano-sol samples prepared in this study.

**Figure 4 membranes-09-00103-f004:**
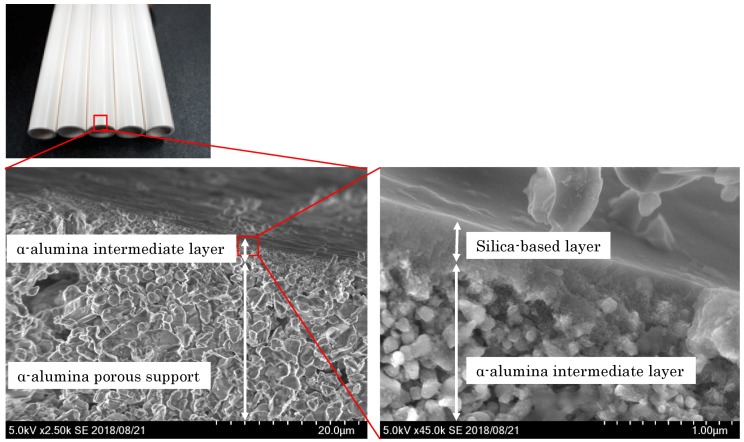
Typical SEM images of the synthesized nano-porous silica-based supports.

**Figure 5 membranes-09-00103-f005:**
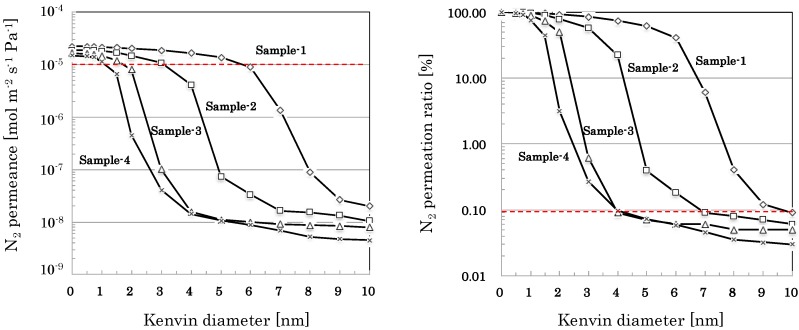
Nano-permporometry profiles of nano-porous supports prepared by coating SiO_2_-based nano-sol samples with different particle sizes. Sample-1, 2, and 3 were prepared by directly coating each of their nano-sol on the α-alumina intermediate layer. Sample-4 was prepared by coating the nano-sol of Sample-4 on the surface of the nano-porous support of Sample-2.

**Figure 6 membranes-09-00103-f006:**
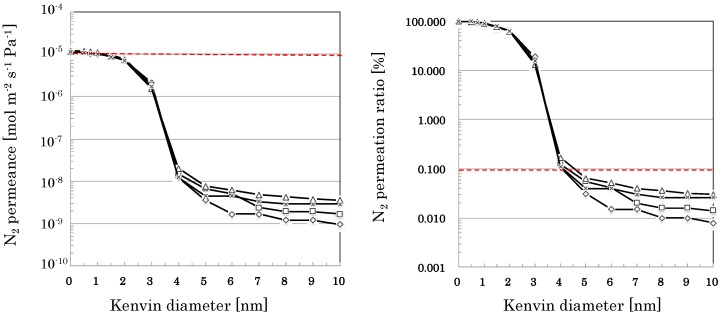
Nano-permporometry profiles of SiO_2_-based nano-porous supports prepared by coating the nano-sol of Sample-3 with the flow-type apparatus.

**Figure 7 membranes-09-00103-f007:**
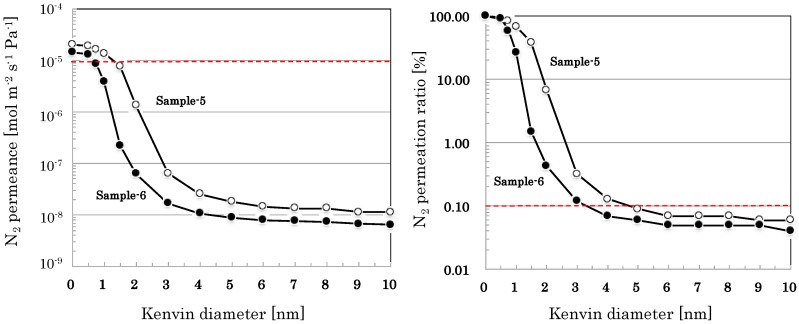
Nano-permporometry profiles of nano-porous supports prepared by coating SiO_2_–ZrO_2_-based nano-sol samples with different particle sizes. Sample-5 was prepared by directly coating the nano-sol Sample-5 on the α-alumina intermediate layer. Sample-6 was prepared by coating the nano-sol of Sample-6 on the surface of the nano-porous support of Sample-5.

**Table 1 membranes-09-00103-t001:** Experimental conditions of nano-permporometry measurements in this study.

Step	P/Ps [-]	Kenvin Diameter [nm]	N_2_ Flow [L(STP) min^−1^]	
			Dry	Wet
1	0	0	5.00	0.00
2	0.015	0.50	4.92	0.08
3	0.062	0.75	4.69	0.31
4	0.124	1.0	4.38	0.62
5	0.248	1.5	3.76	1.24
6	0.352	2.0	3.24	1.76
7	0.498	3.0	2.51	2.49
8	0.593	4.0	2.04	2.97
9	0.658	5.0	1.71	3.29
10	0.706	6.0	1.47	3.53
11	0.742	7.0	1.29	3.71
12	0.770	8.0	1.15	3.85
13	0.793	9.0	1.04	3.97
14	0.811	10.0	0.95	4.06

**Table 2 membranes-09-00103-t002:** List of nano sol samples and their estimated particle sizes.

Nano-Sol	Material	Preparation Conditions	Method	Estimaed Particle Size [nm]
Sample-1	SiO_2_	2 wt % in water, aging at 40 °C for 30 day	DLS, TEM	10–25
Sample-2	SiO_2_	1 wt % in water, aging at 40 °C for 30 day	DLS, TEM	5–15
Sample-3	SiO_2_	1 wt % in water, aging at 25 °C for 30 day	DLS	3–10
Sample-4	SiO_2_	0.5 wt % in water, aging at 25 °C for 7 day	DLS, TEM	2–10
Sample-5	SiO_2_-ZrO_2_	1 wt % in water, agin at 25 °C for 7 day	DLS	2–10
Sample-6	SiO_2_-ZrO_2_	0.5 wt % in water, aging at 25 °C for 7 day	DLS, TEM	2–5
Sample-7	Organo-silica (BTESE)	0.5 wt % in ethanol, aging at 25 °C for 1 day	DLS	1–2

**Table 3 membranes-09-00103-t003:** Pore size distribution of SiO_2_-based nano-porous supports synthesized using different nano-sols.

Permeation Path r [nm]	Permeation Ratio [%]			
	Sample-1	Sample-2	Sample-3	Sample-4
0 < r ≦ 1	2.9	3.0	10.6	23.9
1 < r ≦ 2	5.4	18.3	40.1	73.0
2 < r ≦ 3	7.2	20.8	48.7	2.8
3 < r ≦ 4	10.1	35.7	0.53	0.17
4 < r ≦ 5	13.0	21.7	0.02	0.02
5 < r ≦ 6	20.9	0.21	0.01	0.01
6 < r ≦ 7	34.5	0.09	0.00	0.01
7 < r ≦ 8	5.72	0.01	0.01	0.01
8 < r ≦ 9	0.28	0.01	0.00	0.00
9 < r ≦ 10	0.03	0.01	0.00	0.00
10 < r	0.09	0.06	0.05	0.03

**Table 4 membranes-09-00103-t004:** Pore size distribution of SiO_2_-based nano-porous supports prepared by coating the nano-sol of Sample-3 with the flow-type apparatus.

Permeation Path r [nm]	Permeation Ratio [%]			
	Sample-3 (F1)	Sample-3 (F2)	Sample-3 (F3)	Sample-3 (F4)
0 < r ≦ 1	10.5	10.5	9.3	8.5
1 < r ≦ 2	25.9	25.4	28.7	27.7
2 < r ≦ 3	44.7	48.4	48.7	48.2
3 < r ≦ 4	18.8	15.6	13.2	15.5
4 < r ≦ 5	0.08	0.07	0.11	0.07
5 < r ≦ 6	0.02	0.01	0.01	0.00
6 < r ≦ 7	0.00	0.02	0.01	0.01
7 < r ≦ 8	0.01	0.00	0.00	0.00
8 < r ≦ 9	0.00	0.00	0.00	0.00
9 < r ≦ 10	0.00	0.00	0.00	0.00
10 < r	0.01	0.01	0.03	0.03

**Table 5 membranes-09-00103-t005:** Pore size distribution of SiO_2_–ZrO_2_-based nano-porous supports synthesized using different nano-sols.

Permeation Path r [nm]	Permeation Ratio [%]	
	Sample-5	Sample-6
0 < r ≦ 1	31.8	73.4
1 < r ≦ 2	61.5	26.2
2 < r ≦ 3	6.4	0.31
3 < r ≦ 4	0.19	0.05
4 < r ≦ 5	0.04	0.01
5 < r ≦ 6	0.02	0.01
6 < r ≦ 7	0.00	0.00
7 < r ≦ 8	0.00	0.00
8 < r ≦ 9	0.01	0.00
9 < r ≦ 10	0.00	0.01
10 < r	0.06	0.04

**Table 6 membranes-09-00103-t006:** Gas permeation properties of membranes synthesized using the nano-porous supports of Sample-6 and nano-sol of Sample-7.

Membrane	H_2_ Permeance 10^−6^ [mol m^−2^ s^−1^ Pa^−1^]	Ratio of Permeance [ - ]	
		H_2_/CH_4_	H_2_/SF_6_
Membrane 1	5.21	10.6	>2000
Membrane 2	4.85	19.7	>2000
Membrane 3	4.05	23.7	>2000
Membrane 4	5.60	14.0	>2000
Membrane 5	4.44	21.4	>2000
Membrane 6	5.56	8.2	273

**Table 7 membranes-09-00103-t007:** H_2_ permeation properties of 1,2-bis(triethoxysilyl)ethane (BTESE)-derived silica membranes.

Intermediate Layer			After BTESE-Derived Silica Coated					Reference
Material	N_2_ Permeance [10^−6^ mol m^−2^ s^−1^ Pa^−1^]	Average Pore Size [nm]	H_2_ Permeance [10^−6^ mol m^−2^ s^−1^ Pa^−1^]	Selectivity			Measurement Temperature [°C]	
				H_2_/N_2_	H_2_/CH_4_	H_2_/SF_6_		
γ-alumina	0.1–1	1.7–5	0.2–0.5	7–20	7–20	-	200	ten Hove et al. [12,13]
γ-alumina	-	4	0.25	20–30	-	-	200	Agirre et al. [14]
SiO_2_-ZrO_2_	4	0.65	1	up to 100	-	-	200	Nagasawa et al. [15]
SiO_2_-ZrO_2_	-	several	1–4	10–100	-	100–10,000	100–200	Yu et al. [16]
SiO_2_-ZrO_2_	-	several	2–10	9–23	-	1000–25,500	100–300	Kanezashi et al. [5]
SiO_2_-ZrO_2_	15	0.8	4.4–5.6	-	10–23	>2000	100	This work

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
