# Peer review of "Development of Mass Production Technology of Highly Permeable Nano-Porous Supports for Silica-Based Separation Membranes"

_membranes, 2019, doi:10.3390/membranes9080103_

Round 1

Reviewer 1 Report

The manuscript developed mass producing technologies of nano-porous supports, and the developed mass production apparatus had reproducibly produce nano-porous silica-based supports with high permeance  and minimizied effects of membrane  defects. Furthermore, an explanation of why the authors did these various experiments should be provided.Besides, the authors should give a comparison of the membrane permeance with the references.

Reviewer 2 Report

The authors reported the “Development of Mass production “ of  silica bases supports.

However, the mass production procedure is not explained. In Fig. 1, batch and flow type apparatus are displayed, however, there is not information about the production.  Since the manuscript is related to “mass production”, a detail information should be present. The authors should submit again adding the requested information.

Other suggestions.

2.-  Materials and synthesis.

More detail information of preparation of the colloidal solutions of the different layers. How many times the dip-carbonization cycles were carried out.  Dipping concentration, time of dipping, firing conditions (i.e. soaking time).

The support was symmetric or asymmetric?.

Permeation test.-  Explain the sealings.

It could be interesting to have the H2 permeance at various temperatures.

Pag 8.-  line 196.   What is the evidence that the pore is hydrophobic?

Round 2

Reviewer 2 Report

I am happy with the modifications, I recommend the article to be published

Author Response

Thank you for your valuable comments and suggestions.

We are grateful for your cooperation.